# Metabolic Plasticity of Astrocytes and Aging of the Brain

**DOI:** 10.3390/ijms20040941

**Published:** 2019-02-21

**Authors:** Mitsuhiro Morita, Hiroko Ikeshima-Kataoka, Marko Kreft, Nina Vardjan, Robert Zorec, Mami Noda

**Affiliations:** 1Department of Biology, Graduate School of Sciences, Kobe University, 657-8501 Kobe, Japan; 2Faculty of Science and Engineering, Waseda University, 3-4-1 Okubo, Shinjuku-ku, Tokyo 169-8555, Japan; ikeshima@1988.jukuin.keio.ac.jp; 3Laboratory of Cell Engineering, Celica Biomedical, 1000 Ljubljana, Slovenia; Marko.Kreft@mf.uni-lj.si (M.K.); Nina.Vardjan@mf.uni-lj.si (N.V.); robert.zorec@mf.uni-lj.si (R.Z.); 4Laboratory of Neuroendocrinology—Molecular Cell Physiology, Institute of Pathophysiology, Faculty of Medicine, University of Ljubljana, 1000 Ljubljana, Slovenia; 5Department of Biology, Biotechnical Faculty University of Ljubljana, 1000 Ljubljana, Slovenia; 6Laboratory of Pathophysiology, Graduate School of Pharmaceutical Sciences, Kyushu University, Fukuoka 812-8582, Japan; noda@phar.kyushu-u.ac.jp

**Keywords:** astrocyte, metabolism, glucose, fatty acid, insulin, noradrenaline, thyroid hormone

## Abstract

As part of the blood-brain-barrier, astrocytes are ideally positioned between cerebral vasculature and neuronal synapses to mediate nutrient uptake from the systemic circulation. In addition, astrocytes have a robust enzymatic capacity of glycolysis, glycogenesis and lipid metabolism, managing nutrient support in the brain parenchyma for neuronal consumption. Here, we review the plasticity of astrocyte energy metabolism under physiologic and pathologic conditions, highlighting age-dependent brain dysfunctions. In astrocytes, glycolysis and glycogenesis are regulated by noradrenaline and insulin, respectively, while mitochondrial ATP production and fatty acid oxidation are influenced by the thyroid hormone. These regulations are essential for maintaining normal brain activities, and impairments of these processes may lead to neurodegeneration and cognitive decline. Metabolic plasticity is also associated with (re)activation of astrocytes, a process associated with pathologic events. It is likely that the recently described neurodegenerative and neuroprotective subpopulations of reactive astrocytes metabolize distinct energy substrates, and that this preference is supposed to explain some of their impacts on pathologic processes. Importantly, physiologic and pathologic properties of astrocytic metabolic plasticity bear translational potential in defining new potential diagnostic biomarkers and novel therapeutic targets to mitigate neurodegeneration and age-related brain dysfunctions.

## 1. Introduction: Astrocyte and Brain Energy Metabolism

The human brain represents merely 2% of body mass; however, it consumes approximately 20% of energy substrates at rest, and energy consumption by the brain can be further elevated during various tasks [1,2]. This relatively effective energy handling by the brain depends on the metabolic plasticity of astrocytes, a type of neuroglial cell, abundantly present in the mammalian brain and anatomically positioned between densely packed neuronal structures and the complex ramification of cerebral vasculature [3]. Therefore, astrocytes are structural intermediates between blood vessels and neurons, delivering blood-derived glucose to neurons, which are the main energy consuming elements of the brain, and it is likely that age-dependent or disease-related alterations of astrocytes affect brain homeostasis and activities [3], and may even lead to accelerated pathologic processes under some conditions, including aging.

Together with endothelial cells and pericytes, astrocytes form the blood-brain-barrier (BBB), a structure for transporting various molecules and nutrients, including glucose through the transporter GLUT1 [4], monocarboxylates, such as L-lactate through the monocarboxylate transporter (MCT) [5] and fatty acids through fatty acid translocase (FAT) [6]. These molecules play crucial roles in the exchange of energy substrates between the blood and the brain parenchyma. Thus, the vast activity-dependent neuronal energy consumption, reflecting the maintenance of electrical signaling and stability of intracellular concentration of ions and synaptic vesicle cycling, is supported by astrocytes [7].

It is well established that glucose is an obligatory fuel, critically important for many brain functions, including ATP production, oxidative stress management, and synthesis of neurotransmitters, neuromodulators, and structural components of the cell [2]. However, the delivery of glucose and its metabolites to brain parenchyma is still under debate. The experimentally-determined ratio between glucose and oxygen consumption at rest suggests the incomplete oxidation of glucose due to substantial lipid and/or amino acid production from glucose, or the excretion of unoxidized metabolite, especially L-lactate [8]. The incomplete glucose oxidation, together with L-lactate accumulation after neuronal activity [9], indicates the overwhelming capacity of glycolysis in comparison with oxidative metabolism. The relatively large glycolytic capacity of brain tissue is most likely attributed to astrocytes [1,10], where glycolysis appears to have a larger enzymatic capacity than oxidative metabolism [11], and neuronal glycolysis is limited [12]. In addition, astrocytic glycolysis is boosted by the neurotransmitters glutamate and noradrenaline (NA) [13]. Hence, neuronal ATP production with astrocyte-derived L-lactate was proposed as a model of activity-dependent energy metabolism called astrocyte-neuron L-lactate shuttle (ANLS) [14], and its involvement in cognitive function is experimentally suggested [15,16]. However, this model is criticized by at least the following points, namely, (i) the ANLS is inconsistent with the existing data on stoichiometry of brain metabolism and with the rapid excretion of L-lactate after neuronal activity [17] and (ii) the capacity of neuronal glucose uptake and oxidative metabolism is large enough for maintaining their energy consumption during activities [18]. 

Normal brain activities require the activity-dependent glucose supply from blood, as well as from glycogen stored mainly if not exclusively in astrocytes. The uptake of glutamate increases glycogen levels in astrocytes [19], while the inhibition of glycogenolysis suppresses the uptake of glutamate [20] and potassium [21]. In addition, the glycogen in white matter astrocytes is essential for the activity and survival of axons [22]. Thus, astrocyte glycogen likely fuels some specific activities and extends brain activities, especially the number of neurons involved and duration of activities beyond the limitation of the glucose supply from blood flow. Depletion of astrocytic glycogen is supposed to cause reduced brain activity leading to cognitive dysfunction and neurodegeneration.

Astrocytes are also equipped with lipid metabolism, and the lipids produced by astrocytes are supplied to neurons and oligodendrocytes as components of synaptic and myelin membranes [23,24]. Oxidative metabolism of lipids is limited in comparison with that of glucose in the resting brain [23], but it is also regulated by the interaction between astrocytes and neurons. Ketone bodies are a major energy substrate in neonatal brains [25], and the ketone bodies produced by liver during starvation or in diabetes patients are taken by the brain via MCTs for energy production [26]. The neuroprotection by ketone bodies is well-established and clinically used as high-ketogenic diet for epilepsy and brain injury [27,28]. Astrocytes respond to ischemia in vitro by enhancing ketone body production via AMP-activated protein kinase (AMPK), and astrocyte-produced ketone bodies, derived from fatty acids, might serve as a neuronal energy substrate for the tricarboxylic acid (TCA) cycle instead of L-lactate, as pyruvate dehydrogenase is susceptible to ischemia [29]. Thus, under particular conditions, fatty acids may be used for ketone body production initially by astrocytes, then exchanged between astrocytes and neurons via MCTs for the energy production by the TCA cycle.

In this paper we review the plasticity of astrocytic energy metabolism and its implications in diseases and aging of the brain. Astrocytic energy metabolism is regulated by physiologic and pathologic factors, including hormones, neurotransmitters, cytokines, reactive oxygen species (ROS), ions, oxygen and nutrients, and overall astrocytic metabolic adapting capacity affects neuronal activities via exchange of metabolites between astrocytes and neurons. Furthermore, under conditions where astrocytes become reactive, this adds to the complexity of understanding astrocytic energy metabolism. Accumulating evidence indicates a connection between diseases and aging of the brain and metabolic plasticity of astrocytes, especially reflecting failure of physiologic regulation of metabolic plasticity under pathologic conditions. Thus, revealing the machinery behind the astrocytic metabolic plasticity is expected to lead us to novel strategies for therapeutic treatment of age-related brain diseases, including dementia, and also for maintaining a healthy aging.

## 2. Glycogen Stores in The Brain

To support morphologic plasticity of the brain, which underlies memory formation and cognition [30], relatively large amounts of energy are required. While neurons consume most of the energy, they lack energy stores [31]. In the brain, energy is stored in the form of glycogen, which is localized almost exclusively in astrocytes [32,33]. The steady-state amount of astrocytic glycogen, determined by dynamic equilibrium between synthesis (glycogenesis) and glycogenolysis [34], is relatively low in comparison with that in the skeletal muscle and liver; however, it is thought to contribute importantly to neuronal function [35]. Astrocytic glycogen utilization can support neuronal activity in hypoglycemic conditions and through transient events where neuronal activity is increased [35,36,37]. Glycogenolysis is facilitated by the lack of energy substrates [38] and by elevated neural network activity [39,40], as is the case of sleep deprivation, during which glycogen content is strongly reduced [41]. Hence, glycogen seems to be a short-term energy buffer, importantly bridging transient energy requirements, rather than playing a role in sustained energy provision [42]. In the next two subheadings we discuss how NA and insulin modulate glycogen levels in astrocytes and how this may affect neurologic diseases.

## 3. Adrenergic Regulation of Astrocytic Glycolysis and Metabolic Excitability 

Although the brain, representing only about one or two percent of body mass, consumes one fifth of body glucose at rest [43], this has to be further increased during some conditions, including alertness, awareness and attention. These states require simultaneous, coordinated and spatially homogeneous activity within a large area of the brain neural networks. This is considered to be provided through the noradrenergic neurons, arising from the locus coeruleus (LC) in the brainstem [44].

This relatively small nucleus, named by the Wenzel brothers in 1812, referring to its bluish color [45,46], was described in 1784 by Félix Vicq-d’Azyr (1748–1794) and re-discovered by Johann Christian Reil in 1809 [47]. In humans (19–78 years) the LC has around 50,000 neurons, which are pigmented by neuromelanin, a polymerization product of NA [47,48]. In 1959 it was reported that these neurons exhibit high monoamine oxidase activity. In 1964 monoamines were identified, and in the 1970s the canvas of noradrenergic ubiquitous projections to the central nervous system (CNS) were described [45]. There is consensus in the field that LC is the main source of NA in the CNS [49,50,51], contributing ~70% of all NA in the brain.

The axons of LC neurons ramify widely [50,52,53], and project to most of the CNS areas, including the cerebellum, the spinal cord, the brain stem, the hypothalamus and thalamus, the basal ganglia and the cortex, where some areas may be more abundantly innervated [53]. The structure of this neuronal connectome represents a morphological basis through which synchronous activation, a form of “reset” mechanism [54,55], of many neural networks in the CNS can be elicited. Functionally, this likely mirrors in γ waves on an electroencephalogram [55] and is essential for LC-associated regulation of the sleep–wake cycle, arousal, attention, memory formation, behavioral flexibility, stress, cognition, emotions and neuroplasticity [52].

When NA is released from LC-neurons through multiple axonal varicosities, it binds to α- and β-adrenergic receptors (ARs), which are expressed in brain cells, including astrocytes, highly heterogeneous neuroglial cells, responsible for homeostatic functions in the CNS [3]. Binding of NA to astrocyte α- and β-ARs elicits an increase in cytoplasmic second messengers, Ca^2+^ and cAMP, respectively, which have distinct temporal profiles [56] and trigger many downstream cellular processes including morphologic cell reshaping [57,58] and aerobic glycolysis [59,60].

Why are astrocytes an important target of LC-released NA? One reason for this is that about half of the LC axon terminals do not form classical synapses [49]; therefore, NA released from these varicose terminals escapes into the “volume”, reaching also ARs on astrocytes, especially the β-ARs, which are abundantly expressed by astrocytes in both white and grey matter [61,62,63,64,65]. These receptors are likely to be involved through astrocytic shape changes in memory formation [30,66], an energy requiring process [43]. Indeed, memory consolidation requires glycogenolysis in young chickens [67,68]. The consolidation of memory, from short- to long-term, requires neuronal NA release [69]. Astrocytes that store energy in the form of glycogen in the brain [31], upon activation by NA generate metabolites including L-lactate [1], acting as a fuel for active neurons [70], and also coordinate neighboring astrocytes as a signal of the syncytium to further enhance astrocytic aerobic glycolysis [60]. This new mechanism, termed “metabolic excitability” appears to be affected in age-related neurodegeneration and cognitive decline, which depends on the availability of NA due to the age related demise of LC [71,72]. 

Considering that the LC-dependent deficit drives neurodegeneration, the astrocytic contribution to this pathology is expected to be important, depending on the loss of NA [66,73]. Indeed, during normal aging up to 25% of LC neurons, responsible for ~50% of brain NA levels, are lost in the elderly (>90 years age) [51]. Therefore several strategies have been proposed to prevent neurodegeneration by astrocytic adrenergic activation [73] including the (i) exposure of the subjects to enriched sensory environment or to electrical stimulation, even by deep brain stimulation [74]; (ii) transplanting noradrenergic neurons [66,75]; and (iii) applying drugs that elevate or normalize NA [76] and/or agonists acting in a similar manner as those on ARs, although via different receptors. It is the latter strategy that may be employed to mimic the action of NA under conditions when NA levels are reduced due to the age- and or disease-related demise of LC neurons. To select such drugs, it will be important to test whether they affect glycogenolysis in astrocytes and whether this is similar to that elicited by NA.

## 4. Insulin and Insulin-Like Growth Factor 1 Regulation of Astrocytic Metabolism

When glucose enters astrocytes, a part of available glucose is temporarily converted to glycogen and subsequently released from glycogen. This rapid turnover of glucose through glycogen is termed the “glycogen shunt pathway” [77] and plays an important role in determining cytoplasmic levels of free glucose. 

The steady-state amount of astrocytic glycogen is regulated by a number of mediators [78,79]. Some of these, such as NA, stimulate glycogenolysis. Others, such as insulin, play an important role in glycogenesis. Insulin receptors are expressed in brain cells [80] including astrocytes [81,82]. While insulin production in the brain is considered low or even absent [83,84], insulin can enter the brain via regions with loose BBB, represented by circumventricular organs [85], and by receptor-mediated active transport system [86,87,88,89,90]. Experimental evidences support the view that insulin-mediated signaling is involved in the neuronal survival [91] and the regulation of body food intake [92,93,94], and that it affects cognition and memory [95,96,97,98,99]. Thus, it is also considered that insufficient levels of insulin may contribute to the neurological and psychiatric complications of diabetes [96,100,101,102]. In the CNS, impaired insulin action may aggravate the course of neurodegeneration associated with Alzheimer’s [103] and Parkinson’s disease [104]. Interestingly, compared with control subjects, plasma insulin levels in Alzheimer’s disease patients are higher, whereas insulin levels in the cerebrospinal fluid are lower [105,106]. A double-blind, placebo-controlled, within-subject comparison in 15 healthy men showed, that cerebral insulin suppresses food intake after an overnight fast [107]. 

Insulin also acts as a growth factor and increased levels of insulin promote the growth of rodent astrocytes in cultures [108]. Importantly, insulin addition to astrocytes in culture stimulates the amount of glycogen [36,108,109,110]. While it is not known how insulin regulates glycogen stores in astrocytes, it is considered that insulin stimulation augments glucose uptake into astrocytes [79,110]. In skeletal muscle, insulin enhances the incorporation of the glucose transporter GLUT4 into the sarcolemma; therefore, a similar mechanism may play a role in astrocytes since GLUT4 expression in astrocytes was confirmed with immunocytochemical staining [111,112]. Interestingly, in the study by Muhic et al. [111] where a glucose nanosensor based on Förster resonance energy transfer (FRET), allowing measurements of dynamic changes of glucose concentration in single cells with high temporal resolution, was used [113], it was shown that insulin does not increase the flux of glucose across the plasma membrane, while it does increases levels of cellular glycogen [111].

It has also been considered that insulin signaling in the CNS may be involved in declarative memory formation and that the impaired brain insulin signaling plays a critical role in the loss of memory functions associated with Alzheimer’s disease [114]. A double blind placebo-controlled study on 25 patients showed that intranasal insulin application improved retainment of verbal information after a delay, as well as attention and functional status of patients with Alzheimer’s disease [115]. PET imaging indicated, that ^18^F fluorodeoxyglucose uptake was higher in some regions of the CNS following intranasal insulin administration compared to placebo [98]. Similarly, a recent study determined that four months of treatment with intranasal insulin improves memory; however, treatment with the insulin detemir, a long-acting human insulin analogue, led to no significant effects [105]. Clinical studies thus indicate that insulin signaling affects cognition, likely via astrocytes. The physiological and pathophysiological mechanisms of insulin action in the brain in relation to aging and longevity remain to be elucidated [116].

In addition to insulin, which affects the behavior of animals via the gliotransmitter ATP of astrocytes [96], insulin-like growth factor 1 (IGF-1) may also be important in astrocytic control of CNS metabolism. As insulin, also IGF-1 likely modulates glycogen stores in astrocytes [111]. It was recently reported that a reduction in IGF-1 receptor (IGFR) expression, typically with aging, is associated with decline in hippocampus-dependent learning and increased gliosis [117]. Moreover, astrocyte-specific knockout of IGFR impaired working memory. Furthermore, reducing IGF-1 signaling in primary astrocyte culture by a 50% reduction in IGFR expression significantly impaired ATP synthesis, likely due to altered mitochondrial structure and function leading to ROS production associated with the induction of an antioxidant response [117]. A similar impairment of mitochondrial complex I function was observed in human astrocytes and this affected the survival of human co-cultured neurons [118]. 

## 5. Thyroid Regulation of Brain Metabolic Rate

Brain function is influenced by the endocrine system, especially by thyroid hormones (THs). The glioendocrine system, a term originally proposed to describe interactions between the endocrine system and glial cells [119], further highlights the contribution of glial cells in the endocrine-mediated regulation of the CNS function [120].

Since astrocytes metabolize THs to active form [121], they play a central role in the endocrine control of neural environment [122]. THs have to cross multiple membranes in order to reach their receptors in the nuclei and mitochondria in addition to the ones in the cytoplasm. In particular, THs need to enter the brain through the BBB. Various TH transporters were reported in mouse [123] and human brain [124]. Astrocytes, forming partly the BBB, are the main cell population incorporating circulating L-thyroxine (T4) through TH transporters. Incorporated T4 in astrocytes is deiodinated by the type 2-deiodinase (D2) to produce 3, 3′, 5-triiodothyronine (T3) [125]. Subsequently T3 is released from astrocytes and taken by other cell types via distinct membrane transporters (paracrine signaling). For example, adjacent neurons express TH receptors and type 3-deiodinase (D3), which inactivates T3. Since the neuronal paracrine pathway is regulated by hypoxia, ischemia, or inflammation, it is postulated that deiodinases could act as potential control points for the regulation of TH signaling in the brain during health and disease [126].

Cultured astrocytes express relevant genes of T3 receptors (Thrα1 and ThRβ), presumably both in the nucleus/mitochondria and in the cytoplasm, and nuclear corepressor (Ncor1) and coactivator (Ncoa1) [127], in addition to D2 and TH transporter (Mct8/Slc16a2) (autocrine signaling) [127]. During CNS development, T3 exerts various effects including astrocyte differentiation [128,129], as well as neuronal maturation due to astrocytic production of extracellular matrix proteins and growth factors [121]. 

The prevalence of thyroid disorders increases with age [130]. In particular, the prevalence of subclinical hypothyroidism ranges from 3 to 16% in individuals aged 60 years and older [131]. Abnormal levels of THs often cause psychological and behavioral abnormality. In addition, hypothyroidism is one of the most common causes of cognitive impairment [132,133,134] and can lead to psychiatric symptoms [135]. On the other hand, ironically and interestingly decreased thyroid function may lead to extended longevity [136].

Mitochondrial metabolism in astrocytes plays a significant role in neuroprotection. Mitochondrial energy production is rapidly increased via a mitochondrial targeted TH receptor after treatment with T3 [137]. Therefore, targeting astrocyte metabolism to increase brain ATP levels could be an efficient strategy to enhance neuroprotection. Another stimulant of mitochondrial ATP production in astrocyte are P2Y1 receptor agonists, which show neuroprotective effect after cerebral ischemic stroke [138,139]. Stimulating endogenous ATP release from astrocytes has also been reported to induce antidepressant-like effects in mouse models of depression [140]. 

Although most energy in the CNS is derived from glucose catabolism, significant energy can also be derived from fatty acid oxidation (FAO) which is stimulated by THs. Studies in rat brains revealed that FAO contributes approximately 20% of brain oxidative energy production [141]. It has been shown that hydroxyacyl-CoA dehydrogenase/3-ketoacyl-CoA thiolase/enoyl-CoA hydratase alpha (HADHA), an essential component of the mitochondrial trifunctional protein complex in the FAO cycle, is critical for the FAO regulation by T3 [142]. Since 95% of HADHA co-localize with glial-fibrillary acidic protein (GFAP) in the brain, T3 is considered to upregulate HADHA and subsequent neuroprotective mitochondrial energy production via FAO in astrocytes. Indeed, T3-treatment decreased stroke volumes in mice, while T3 had no protective effect on stroke volume in HADHA +/− mice [143]. 

Analyses of human brain gene expression databases indicated that the Chromosome 12p12 locus, where many genomic markers are related to dementia risk, may regulate particular astrocyte-expressed genes induced by T3. Two single nucleotide polymorphisms (SNPs) on chromosome 12p12, rs704180 and rs73069071, were found as risk alleles for non-Alzheimer’s neurodegeneration [144,145] and hippocampal sclerosis pathology [146]. The rs73069071 risk genotype is also associated with altered expression of a nearby astrocyte-expressed gene, *SLCO1C1*. SLCO1C1 protein transports TH into astrocytes from blood. Interestingly, total T3 levels in cerebrospinal fluid (CSF) are elevated in hippocampal sclerosis cases but not in Alzheimer’s disease cases, relative to controls [146]. This suggests that even normal levels of T3 in the CSF, astrocyte-TH dysregulation in the brain due to genetic modification contribute to dementia in the elderly. Energy metabolism in hypothyroid brain leads to disruption in astrocyte cytoskeleton as well as glutamatergic and cholinergic neurotransmission, Ca^2+^ equilibrium, redox balance, morphological and functional aspects in the cerebral cortex even in young rats from maternal hypothyroidism [147].

## 6. Other Pathways Regulating Astrocytic Lipid Metabolism

One of the major signals regulating the lipid metabolism in adipocytes is a nuclear receptor, Peroxisome Proliferator-Activated Receptor gamma (PPARγ), which is also expressed in astrocytes [148]. A PPARγ agonist, thiazolidinedione is shown to upregulate the glucose uptake by astrocytes, as known in liver, adipose tissue and muscle; however, this is not mediated by transcriptional changes via PPARγ [149]. The same non-transcriptional action of PPARγ agonist may underlie the rapid inactivation of reactive astrocytes, namely, the reduction of GFAP in reactive astrocytes after spinal cord injury within one hour of treatment [150], even though the anti-inflammatory action of PPARγ agonist on microglia [151] cannot be excluded. Astrocyte selective ablation of PPARγ causes development of diabetes-related symptom, including leptin or glucose resistances [148]. The pathway by which astrocytes PPARγ regulates peripheral metabolism is still to be determined. However, the upregulation of astrocyte glutamate transporter by PPARγ agonist [152] is supposed to change feeding behavior by affecting the activities in the hypothalamic feeding center.

Ciliary neurotrophic factor (CNTF) up-regulates FAO in astrocytes both in vitro and in vivo [153]. The astrocyte-selective CNTF expression induces hypertrophic morphology of astrocytes and downregulates tissue glycolytic activity. In addition, CNTF treatment induces astrocytic increase in β-oxidation enzymes, while it reduces glycolytic enzymes. Since the main signaling pathway of CNTF involves signal transducer and activator of transcription 3 (STAT3), which is commonly upregulated during astrocyte activation [154], FAO may be a common property of reactive astrocytes. However, the upregulation of FAO by other cytokines is not established.

## 7. Metabolic Plasticity of Reactive Astrocytes

Based on the gene expression profile, two reactive astrocyte subpopulations, neurodegenerative A1 and neuroprotective A2 types, were recently proposed [155]. Neurodegenerative A1 reactive astrocytes are induced by inflammatory factors produced by lipopolysaccharide-treated microglia, and are also present with aging [156]. No alteration of glycolytic gene expression is reported in transcriptome analysis of reactive astrocytes induced in cell culture [155], in animal models [157] or in aging [158]. Meanwhile, pro-inflammatory cytokines were shown to upregulate astrocytic glycogenolysis and glucose oxidation [159]. In addition, nitric oxide produced by microglia, increased the expression of glycolytic genes via hypoxia-inducible factor 1 alpha (HIF1α [160], and activated glycolytic enzymes by AMPK [12]. The discrepancy between transcriptome and biochemical data may reflect the different sensitivity of methods or the complexity of A1 reactive astrocyte, which is induced by combinations of pro-inflammatory factors. The upregulation of astrocytic glycolysis in response to inflammation or oxidative stress produces ATP and L-lactate, which compensates astrocytic oxidative metabolism damaged by nitric oxide, and the supply of energy substrate to neurons, respectively [161]. This glycolysis is considered as a double-edged sword, because it supports neuronal energy metabolism, but accelerates inflammation by fueling neurodegenerative A1 reactive astrocytes.

The upregulation of lipid metabolism in reactive astrocytes after stroke [157], which shares a common gene expression profile with the neuroprotective A2 reactive astrocytes [155], is consistent with the upregulation of fatty acid transporter, CD36 in proliferative and scar-forming reactive astrocytes [162]. The excess of fatty acids deriving from dead and dying cells induces inflammation, not via CD36, but via toll-like receptor [163]. Thus, CD36 is likely involved in the uptake of fatty acids for metabolic consumption, rather than in detecting fatty acids as an inflammatory signal. The proliferative reactive astrocytes forming scar after injury are not necessarily induced by CNTF, which upregulates FAO, because CNTF does not induce astrocyte proliferation [153]. The upregulation of FAO after brain injury is neuroprotective in two ways. First, it attenuates inflammation by clearing fatty acid from degenerating tissue. Second, ketone bodies, produced by astrocytic lipid metabolism, maintain neuronal energy metabolism, a compensation for the glucose metabolism, damaged by the inhibition of pyruvate dehydrogenase complex in the initial step of glucose oxidation by ROS [164] and ischemic conditions [29]. Thus, astrocytic lipid metabolism appears to be important in pathologic condition by compensating the impairment of neuronal glucose energy metabolism.

## 8. Conclusions

As summarized in Figure 1, astrocyte energy metabolism is regulated by hormones including NA, insulin and the thyroid hormone, and influenced by pathology states leading to reactive astrocytes. Hormonal regulations of energy metabolism are crucial for tissues of high energy demand, especially the brain and muscle. Myocytes are a large structure equipped with all machinery for energy metabolism, including glucose and fatty acid oxidation, glycogen storage, and hormonal regulation. Meanwhile, the mammalian brain is packed with tiny neurons bearing fine excitable processes. The large astrocytic capacity of handling energy substrates and its regulation by physiologic and pathologic factors most likely complements the neuronal network, highly specialized for information processing by electrical activity. Thus, normal brain activities largely depend on astrocyte metabolic plasticity, and even a small impairment of astrocyte metabolism may cause a significant decline of brain function. Furthermore, the energy metabolism of reactive astrocytes can be a determinant of pathologic processes. Therefore, recent technology for monitoring brain metabolism, such as positron emission tomography of glucose consumption and spectrum magnetic resonance imaging for L-lactate production, largely reflects astrocyte metabolism and provides valuable information on the health and disease states of the brain. Pharmacological manipulations of astrocyte metabolism are a potent therapeutic strategy for cognitive decline and neurodegeneration.

## Figures and Tables

**Figure 1 ijms-20-00941-f001:**
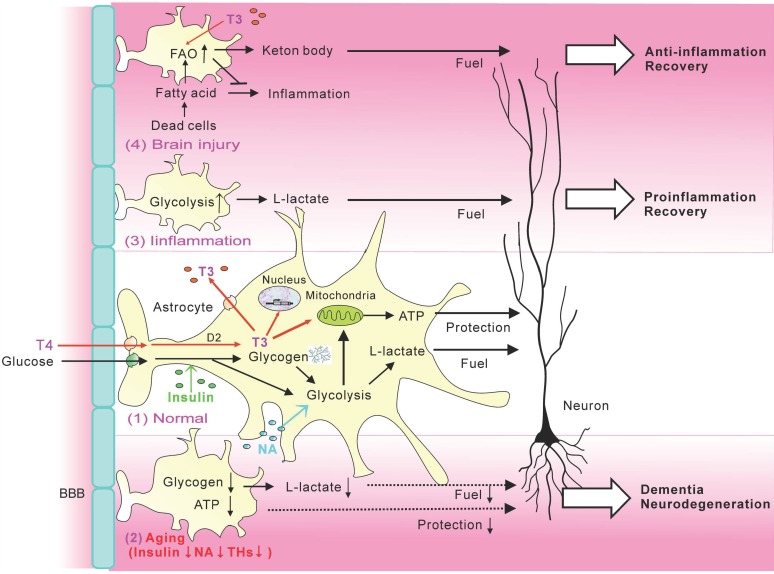
Metabolic plasticity of astrocytes. (1) In normal astrocytes, insulin upregulates glucose uptake from blood stream, NA-stimulated glycolysis, and T4 is converted to T3 and upregulated ATP production in mitochondria. Astrocytic L-lactate fuels neurons, while neuroprotective functions of astrocytes are maintained by ATP production. (2) The reductions of insulin, NA, and THs with aging downregulate astrocytic fuel provision and protection of neurons. (3) Astrocytic glycolysis is upregulated after inflammation, resulting in the increase of fuel for neurons, as well as proinflammatory reactive astrocytes. (4) Astrocytic FAO is upregulated after FAO and further accelerated by T3, resulting in the clearance of inflammatory fatty acid derived from dead cells, as well as the production of ketone bodies for fueling neurons.

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
