# Peer review of "Metabolic Plasticity of Astrocytes and Aging of the Brain"

_ijms, 2019, doi:10.3390/ijms20040941_

Round 1
Reviewer 1 Report
This is a comprehensive review on the physiology and metabolism of astrocytes, as well as on the association of astrocyte homeostasis with neurodegenerative disorders, which could reveal some novel theraputic targets.
The review is well structured and informative.
I only have some minor points (please refer to the uploaded file).

Author Response
We appreciate your favorable and instructive comments The manuscript has been revised in accordance with their suggestions, which we believe have strengthened this manuscript. Our responses to the comments are outlined below, after “>”, and all changes to the previous version are underlinedin the revised version. We hope that this manuscript is now acceptable for publication in International Journal of Molecular Sciences.
> Line 2: All “ageing” were changed to “aging”.
> Line 142: NA was defined as noradrenaline in the introduction and all other “noradrenaline” were changed to “NA”.
> Line 251: ROS was also defined in the introduction and all changed.
> Line 341: LPS was changed to lipopolysaccharide.
> Line 343: This LPS was changed to lipopolysaccharide.
> Line 344: “as a parallel of aging” was changed to “with aging”.
> Line 348: “the nitric oxide” was changed to “nitric oxide”.
> The articles and other grammatical points in the following lines were fixed as suggested; Line 69, 108, 262, 274, 281, 292, 299, 319, 324, 331, 332, 341, 359, 360, 361, 375, 386.
Reviewer 2 Report
In the review “Metabolic plasticity of astrocytes and aging of the brain” by Morita et al, the authors focus on energy metabolism plasticity in astrocytes. The authors describe the different metabolic pathways of astrocytes and how they are dynamically modulated by hormones (insulin, noradrenaline and thyroid hormone) to meet the brain energy demand. Then, they highlight how this metabolic flexibility can be modified in pathological conditions and ageing.
This is a new, important and relevant topic. The review is well organized and clearly described. There are appropriate and adequate references to related works. I have only a comment/request.
The "Insulin-like growth factor-1 regulation of astrocytic metabolism" section focuses primarily on insulin action on brain function rather than on insulin regulation of astrocyte/brain metabolism.
I do not completely agree with the sentence “However, whether insulin signaling is altered in ageing, remains to be verified” (pag 5, row 242). Important references on this topic can be found in the following papers and information about modification of insulin signaling in ageing should be included in the text.
1. Frolich L, et al. Brain insulin and insulin receptors in aging and sporadic Alzheimer’s disease. J Neural Transm 1998;105:423–438.
2. Kullmann S, et al. Brain Insulin Resistance at the Crossroads of Metabolic and Cognitive Disorders in Humans. Physiol Rev. 2016;96:1169–1209
3. Akintola AA, van Heemst D. Insulin, aging, and the brain: mechanisms and implications. Front Endocrinol 2015;6:13.
Minor points
- Figure. Although the authors described through the text and in “conclusions section” the figure, a figure legend that guides the readers through the main results would be useful.
- Pag 8, row 378: “Myocyte is a large structure…” Myocytes or astrocytes
Author Response
We appreciate your favorable and instructive comments The manuscript has been revised in accordance with their suggestions, which we believe have strengthened this manuscript. Our responses to the comments are outlined below, after “>”, and all changes to the previous version are underlined in the revised version. We hope that this manuscript is now acceptable for publication in International Journal of Molecular Sciences.
Major point
I do not completely agree with the sentence “However, whether insulin signaling is altered in ageing, remains to be verified” (pag 5, row 242). Important references on this topic can be found in the following papers and information about modification of insulin signaling in ageing should be included in the text.
> The sentence was changed and one of the paper suggested was cited.
Minor points
- Figure. Although the authors described through the text and in “conclusions section” the figure, a figure legend that guides the readers through the main results would be useful.
> Figure legend was added for describing the energy metabolisms in normal, aging ,inflammatory and injured astrocytes.
- Pag 8, row 378: “Myocyte is a large structure…” Myocytes or astrocytes
> “Myocyte” was changed to “Myocytes”.
Reviewer 3 Report
The review by Mitsuhiro Morita et al, described physiologic pathology and pathologic properties of astrocytic metabolic plasticity and how it can be considered as potential diagnostic biomarkers and novel therapeutic targets to mitigate neurodegeneration and age-related brain dysfunction.
The review is clearly written, it's original and of interest in its field.
I recommend that the review be accepted without revision.
Author Response
We appreciate your recommendation. We hope that this manuscript is now acceptable for publication in International Journal of Molecular Sciences.